# Systematically uncovering the absorbed effective substances of Radix Scutellaria-licorice drug pair in rat plasma against COVID-19 using a combined UHPLC-Q-TOF-MS analysis and target network pharmacology

**Xuqing Wen[1]☯, Weiwei Xie[1]☯, Juan Gao[1], Dedong Zhang[1], Mengxin Yang[2], Zhiqing Zhang[1], Yingfeng Du[2]\*, Yiran Jin[1]\***

**1** Department of Pharmacy, The Second Hospital of Hebei Medical University, Shijiazhuang, Hebei, 050000, P. R. China, **2** Department of Pharmaceutical Analysis, School of Pharmacy, Hebei Medical University, Shijiazhuang, Hebei, 050017, P. R. China

☯ These authors contributed equally to this work.

\* yingfengdu@hotmail.com (YD); jinyiran@sohu.com (YJ)

**Data Availability Statement:** All relevant data are within the paper.

## Abstract

Radix Scutellaria-Licorice drug pair (RSLDP), a frequently used herbal pair with the effect of clearing heat and detoxifying, is the commonly employed drug pair in TCM prescriptions for the treatment of COVID-19. Until now, the metabolism feature and anti-COVID-19 mechanism of RSLDP have not been fully elucidated. In this study, a sensitive and rapid method was developed for the separation and identification of the absorbed constituents of RSLDP in the rat plasma by UHPLC-QTOF-MS. Additionally, we optimized the conventional methodologies of network pharmacology and proposed a new concept called target network pharmacology (T-NP). It used the absorbed constituents and the corresponding targets to generate a compound-target network, and compared to conventional network pharmacology, it could reduce false-positive results. A total of 85 absorbed constituents were identified or tentatively characterized in dosed plasma, including 32 components in the group of Radix Scutellaria, 27 components in the group of Licorice, and 65 components in the group of RSLDP. The results showed that the compatibility of Radix Scutellaria and Licorice increased the number of components in vivo. We found that 106 potential targets among the 61 active compounds in RSLDP were related to COVID-19. And 12 targets (STAT3, AKT1, EGFR, HSP9AA1, MAPK3, JUN, IL6, VEGFA, TNF, IL2, RELA, and STAT1) could be core targets for RSLDP in treating COVID-19. Results from these targets indicate that RSLDP treatment of COVID-19 mainly involves response to chemical stress, response to oxygenates, positive regulation of cytokines, PI3K-Akt signaling pathway, AGE-RAGE signaling pathway for diabetic complications, virus-related pathways such as novel coronavirus and human cytomegalovirus infection, inflammatory immune-related pathways, and so on. The metabolism feature of RSLDP in vivo was systematically uncovered. The combined use of the T-NP method could discover potential drug targets and disclose the biological processes

**Funding:** This study was financially supported by Hebei Administration of Traditional Chinese Medicine (No. 2021133) and Hebei Natural Science Foundation of PR China (No. H2022206384).

**Competing interests:** The authors declare no conflict of interest.

of RSLDP, which will clarify the potential mechanisms of RSLDP in the treatment of COVID-19.

## 1. Introduction

COVID-19 (Coronavirus Disease-2019), caused by severe acute respiratory syndrome corona-virus 2, has swept across China and other countries in the entire world since late December 2019, causing immense losses to human life and property [1]. According to the WHO, there are no approved specific antiviral drugs to defeat COVID-19. In China, the administrations of TCM and health committees in various provinces, with numerous TCM experts, have developed Chinese medicine prevention and treatment programs for COVID-19, most of which are Chinese herbs for clearing heat and removing toxin [2]. And according to the TCM theory, the main treatment tent is to clear heat, remove toxin, resolve dampness, and reinforce healthy qi [3]. The combination of Radix Scutellaria and Licorice drug pair (RSLDP), a frequently used herbal pair with the effect of clearing heat and detoxifying, is the most commonly employed drug pair in Chinese medicine prescriptions for the treatment of COVID-19 [4]. Therefore, traditional Chinese medicine (TCM), especially RSLDP, is playing an important role in the treatment of COVID-19.

Licorice, regarded as the "excellent Chinese herb", could assist herb synergy. As the most important herb used in traditional Chinese medicine, licorice is the first of 179 herbs for treating COVID-19 [2]. Based on the Chinese Pharmacopeia, licorice has the capacity to clear heat, detoxify, replenish qi, and relieve cough [5]. A great number of studies have verified many modern pharmacological activities of licorice, such as antiviral [6], antimicrobial [6], anti-inflammatory [7], anticancer [8], anti-diabetic [9], liver protection [10], expectorant [11], and so on. Licorice contains over 20 triterpenoids and nearly 300 flavonoids, among which glycyrrhizic acid, glycyrrhetic acid, and isoliquiritigenin are the main active components [12]. Recent studies have suggested that licorice has anti-COVID properties [13,14]. Radix Scutellaria, which is the dried root of Scutellaria baicalensis Georgi, a multi-year herb of the Lamiaceae, is one of the most frequently used traditional Chinese medicines. Studies demonstrate that Radix Scutellaria possesses extensive pharmacological activities, including anti-inflammatory, antiviral, and anticancer [15]. Flavonoids are the most abundant constituents, among which wogonin and wogonoside are the main active constituents [16]. As the popular herb used in traditional Chinese medicine, Radix Scutellaria is the second of 166 herbal formulae for treating COVID-19 [2]. Currently, the study of the RSLDP formulation, Qing-Fei-Pai-Du decoction, on the prevention and treatment of COVID-19 has become the hot pot in TCM research. Therefore, RSLDP played a vital role in treating COVID-19. However, there are fewer studies on the compatibility of RSLDP. And previous RSLDP studies have focused on the content of chemical compositions in vitro [17].

Nowadays, ultra-high performance liquid chromatography coupled with quadrupole time-of-flight mass spectrometry (UHPLC-QTOF-MS), since its combining great advantages of UHPLC and QTOF-MS, has emerged as a powerful tool to identify chemical components in TCMs [18]. So, in this work, we use it for the characterization of absorbed prototype constituents and the metabolites of RSLDP. Recent clinical treatment experience has shown that the bioactivity of RSLDP action mechanism is likely to be the synergistic result of multi-ingredients, multi-targets and multi-pathways to show the advantages. So, we applied the approach of network pharmacology in this study. Recently, network pharmacology, a promising approach

to TCM research, has provided insight into the pharmacological mechanisms of drug action from the standpoint of macroscopic or holistic regulation among drugs, targets, and diseases, which aligns with the overall view of TCM [19]. Often, the absorbed ingredients are assumed to be effective, exerting their therapeutic effects in vivo [20]. However, the drawbacks of traditional network pharmacology, such as plenty of targets but low bioavailability of substances, lead to false-positive results. Therefore, we introduce target network pharmacology (T-NP), which is used to discover the potential therapeutic effects of absorbed components [21]. Therefore, we attempted to use the T-NP approach to analyze and clarify the pharmacological mechanisms of RSLDP. This is, to our knowledge, the first time to investigate the comprehensive results of the absorbed effective substances, the metabolic pathways, and the target pharmacological network analysis of RSLDP in vivo for the treatment of COVID-19.

## 2. Experimental section

### 2.1. Reagents and materials

Acetonitrile, methanol, and formic acid were of LC/MS grade obtained from the Tedia Company (USA). The ultrapure water used for the mobile phase was purchased from Wahaha Group Co., Ltd. (Hangzhou, China). The other reagents were of analytical grade. Wogonoside (wkq21031811), wogonin (wkq21022605), isoliquiritigenin (wkq21050610), Glycyrrhizic acid (wkq21012606), and Glycyrrhetinic acid (wkq21050707) were all supplied by the Sichuan Weikeqi Biological Technology Co., Ltd (Sichuan, PR China). The purity of the five substances was more than 98%. Radix Scutellariae and licorice were purchased from Bozhou (Anhui province, China). All the herbs were identified by Professor Jianhua Wang and kept in the Department of Pharmaceutical Analysis, School of Pharmacy, Hebei Medical University in Shijiazhuang, China.

### 2.2. Preparation of RS, licorice and RSLDP

Radix Scutellariae (20 g) and licorice (20 g) were mixed with 1:1 ratio, then immersed in water (1:10, w/v) for 0.5 h and refluxed for 1.5 h. The aqueous extract was filtered, and the dregs were refluxed again with water (1:8, w/v) under the same conditions. The two extraction solutions were combined and concentrated to 0.6 g mL$^{-1}$. The resulting extract was centrifuged at 13000 rpm for 10 min, then filtered through a 0.22 μm membrane for UHPLC-QTOF-MS analysis. The treated Radix Scutellariae (20 g) and licorice (20 g) alone were processed in the same way.

### 2.3. Animals and drug administration

Twelve male Sprague-Dawley (SD) rats (weights 220±20 g) were provided by the Laboratory Animal Center of Hebei Medical University (Shijiazhuang, China, License number: SCXK (Liao) 2020–0001). The rats were housed in controlled environmental conditions of 20±2°C with 12 h light/dark cycles for one week prior to the experiment. Rats were fasted with free access to water for 12 h ahead of the experiment. All the experimental procedures were in accordance with the guidelines approved by the Animal Facility Guidelines of Hebei Medical University.

All the rats were randomly divided into four groups (n = 3): Group 1, the RS group; Group 2, the Licorice group; Group 3, the RSLDP group; and Group 4, the blank group. The herb extracts (6 g/kg) were orally administered to the corresponding experimental group. The rats in the blank group were given physiological saline at the same volume.

## 2.4. Plasma collection and preparation

The blood samples were collected in a 5 mL eppendorf tube with 1% heparin sodium via the orbital venous plexus from each rat at 0.25, 0.5, 1, 2, 4, 8, 12, and 24 h after oral administration and were centrifuged at 3500 rpm at 4°C for 10 min to obtain plasma. Plasma samples from the different time points in the same group were combined into one sample and stored at -80°C for further analysis. 500 μL plasma was treated with 1500 μL methanol, vortexed for 5 min and centrifuged at 13000 rpm for 10 min at 4°C to remove the protein. Each sample was independently extracted three times. Next, the mixed supernatant was evaporated to dryness under $N_2$ flow at 37°C, and residues were reconstituted in 500 μL solution of acetonitrile and centrifuged at 13000 rpm for 10 min. Finally, 5 μL of supernatant was used for UHPLC-Q-TOF-MS analysis.

## 2.5. UHPLC-QTOF-MS conditions

The UHPLC-QTOF-MS/MS consisted of a Prominence™ UHPLC System (Shimadzu, Japan) and a Triple TOF™ 5600+ system (ABSCIEX, USA) equipped with Duo-Spray™ ion sources in the electrospray ionization (ESI) technology. Chromatographic separation was performed for a $KinetexC_{18}$ (100 mm×3.0 mm, 2.6 μm) column with a Security Guard UHPLC $C_{18}$ column (4.0 mm×3.0 mm, 2.6 μm) from Phenomenex. The mobile phase consisted of water with 0.1% formic acid and acetonitrile with a flow rate of 0.3 ml/min. The gradient elution condition was set to: 5–8% B from 1–2 min, 8–35% B from 2–5 min, 35–64% B from 5–15 min, and 64–95% B from 15–25 min. The 95% solvent B was kept constant for the following 3 min to achieve column balance. The injection volume was 5 μL, and the column temperature was 40°C.

The mass detection was operated on a Triple TOF 5600 + system (AB Sciex, Redwood, CA, USA) with Duo-Spray ion sources operating in the negative. The optimized mass spectrometer parameters were used as follows: -4.5 kV ion spray voltage, 550°C turbo spray temperature, and 20 eV collision energy spread (CES). 35 psi curtain gas, 55 psi nebulizer gas (GS1), 55 psi Heater gas (GS2)—all the gas was nitrogen. The collision energy (CE) was set at -40 eV, and the declustering potential (DP) was -80 eV. The scan range was operated with the mass m/z 100 to m/z 1000.

## 2.6. Data analysis strategy

The absorbed prototype constituents were identified by Peakview 2.1 software. The group 4 (blank group) and herb solution were employed as negative control and positive controls, respectively, and the extracted ion peaks which detected in the dosed groups and herb solution but not in the blank group were identified as prototype compounds. Prototype compounds could be analyzed by retention time, accurate mass, and MS fragmentation patterns. We used a three-step strategy to identify metabolites in rat plasma as follows: (1) The bulk data-mining tools including XIC, MDF, PIF, and NLF of MetabolitePilot 2.0 software were employed for data post-processing. (2) Peakview 2.1 software was used to confirm metabolite structures. (3) Clog P, calculated by the software ChemDraw 19.0, was adopted to distinguish isomers. Usually, the value of Clog P is inversely related to retention time of the metabolites.

## 2.7 Target network pharmacology

**2.7.1. Compound targets of RSLDP.** Since the absorbed constituents in the plasma were more probably the bioactive compounds, the prototype components and metabolites identified by UHPLC-QTOF-MS were employed to construct the chemical information database of RSLDP for network pharmacology research. The prototype components were searched in the

PubChem database and the metabolites were drawn by ChemDraw to obtain their molecular information including molecular formula and canonical SMILES. Then, the canonical SMILES were sent to the Swiss Target Prediction (http://www.swisstargetprediction.ch/), after which the species was set to "Homo sapiens".

**2.7.2. COVID-19 targets.** The COVID-19 targets were collected from five databases, namely, GeneCards (https://www.genecards.org/), OMIM (https://omim.org/), PharmGkb (https://www.pharmgkb.org/), TTD (http://db.idrblab.net/ttd/), and Drugbank (https://go.drugbank.com/%22). The keywords "Novel coronavirus pneumonia", "COVID-19", etc. were used and only "Homo sapiens" was selected.

**2.7.3. Protein-protein interactions (PPIs).** The targets of active ingredients in RSLDP and COVID-19-related targets were used the Bioconductor (R) software (https://bioconductor.org/, version 3.15, released on April 27, 2022) to obtain a Venn diagram and obtain the intersection of drug-related targets and disease-related targets. The above common targets were introduced into the STRING (https://cn.string-db.org/, version 11.5) database. For the newly constructed PPIs network, the condition was limited to "Homo sapiens" with high confidence (combine score > 0.90).

**2.7.4. Network construction.** Network construction was used by the Cytoscape (version 3.9.1) software as follows: (1) Compounds-targets network; (2) PPIs network. Moreover, CytoNCA, a Cytoscape plugin, was applied to calculate the topological properties of the PPIs network to screen core targets based on Betweenness centrality (BC), Degree Centrality (DC), Closeness centrality (CC), and Eigenvector (EC).

**2.7.5. GO and KEGG enrichment analyses.** The enrichment of gene ontology (GO) terms in the biological process, cellular component and molecular function categories, and Kyoto Encyclopedia of Genes and Genomes (KEGG) pathway were analyzed with R software. P value < 0.05 were selected for further analysis.

# 3. Results and discussion

In this study, a total of RS-related compounds, including 22 prototype compounds and 10 metabolites, were identified in group 1; a total of Licorice-related compounds, including 22 prototype compounds and 5 metabolites, were identified in group 2; and a total of RSL-related compounds, including 55 prototype compounds and 10 metabolites, were identified in group 3. The MS data are listed as Tables 1 and 2, and the structures of these compounds are shown in Fig 1. The total ion chromatogram of prototype components and metabolites is shown in Fig 2.

## 3.1 Identification of prototypes of RS, Licorice and RSLDP in rat plasma

We detected a total of 62 prototype compounds in all dosed groups, among which 48 were flavonoids (peaks 5–52), 9 were triterpenoid saponins (peaks 54–62), and the others were phenylpropanoids (peaks 4, 53) or phenolic compounds (peaks 1–3). The results suggested that flavonoids and triterpenoid saponins are the major bioactive components in rat plasma.

**3.1.1. Identification of flavonoids related prototype compounds.** A total of 48 flavonoids were identified in rat plasma, containing 3 flavone C-glycosides, 12 flavone O-glycosides, 12 Free flavones, 5 Chalcones, 6 flavanones, 8 isoflavones, 1 flavonol, and 1 flavanonol. Typically, flavonoids are considered to have a strong response in the negative ion mode. Fragmentation with loss of CO, $H_2O$, CHO, $CH_3$, and the residues of glucose and glucuronic acid and RDA cleavage in MS/MS spectra were employed to identify the structure of flavonoids.

*3.1.1.1 Flavone C-glycosides.* Three Flavone C-glycosides were characterized (peaks 34, 47, and 48). $[M-H-60]^-$, $[M-H-90]^-$ and $[M-H-120]^-$ are the obvious characteristic fragment ion of

**Table 1. Characterization of the absorbed prototype constituents in dosed plasma by UHPLC-QTOF-MS.**

| No. | Formula | [M −H]⁻ $m/z$ | Error (ppm) | $T_R$ (min) | MS/MS fragments | Identification | Clog P | Source |
|---|---|---|---|---|---|---|---|---|
| 1 | $C_7H_6O_3$ | 137.0244 | 2.5 | 7.55 | 93.0345,78.9189,65.0407 | 4-Hydroxybenzoic | - | RSLDP |
| 2 | $C_7H_6O_3$ | 137.0248 | 2.5 | 7.56 | 93.0345,78.9189,65.0407 | Salicylic acid | - | L, RSLDP |
| 3 | $C_7H_6O_4$ | 153.0196 | 1.7 | 6.08 | 153.0197,135.0056,109.0305 | Protocatechuic acid | - | RSLDP |
| 4 | $C_9H_6O_4$ | 177.0195 | 1 | 5.8 | 177.0203.149.0259,133.0302,105.0356 | Esculetin | - | L, RSLDP |
| 5 | $C_{15}H_{10}O_4$ | 253.0504 | -0.9 | 11.68 | 253.0514,209.0610,143.0523,165.028 | Chrysin | - | RSLDP |
| 6 | $C_{15}H_{10}O_4$ | 253.0505 | -0.5 | 6.57 | 224.0495,209.0624,197.0599,135.0090,117.0349 | Daidzein | - | RSLDP |
| 7 | $C_{15}H_{10}O_5$ | 269.0456 | 0.1 | 9.44 | 269.0449,241.0500,223.0383,195.0451,139.0026 | Baicalein | 3 | RS, RSLDP |
| 8 | $C_{15}H_{10}O_5$ | 269.0452 | -0.1 | 9.2 | 251.0368,241.0526,225.0543,197.0608,171.0436 | Norwogonin | 2.93 | RS, RSLDP |
| 9 | $C_{15}H_{10}O_5$ | 269.0454 | -0.1 | 7.56 | 269.0448,241.0576,225.0578,151.0561 | Apigenin | 2.9 | RS, RSLDP |
| 10 | $C_{15}H_{10}O_5$ | 269.0449 | -2.6 | 7.24 | 269.0462,241.0525,225.0578,197.0621 | 5,7,2'-Trihydroxy-flavone | 2.6 | RSL |
| 11 | $C_{15}H_{12}O_4$ | 255.0661 | -0.7 | 7.95 | 255.0675,213.0569,135.0095,119.0507,91.0200 | Isoliquiritigenin | - | RS, L, RSLDP |
| 12 | $C_{15}H_{12}O_5$ | 271.0606 | -2.3 | 8.78 | 271.0616,151.0040,119.0525 107.0141 | Naringenin | - | RSLDP |
| 13 | $C_{15}H_{12}O_6$ | 287.0557 | 1.5 | 7.82 | 161.0236, 135.0440,125.0244 | Carthamidin | - | RSLDP |
| 14 | $C_{16}H_{12}O_4$ | 267.0662 | -0.4 | 10.01 | 252.0381,223.0437,195.0466,135.0080 | Formononetin | - | RSLDP |
| 15 | $C_{16}H_{12}O_5$ | 283.0611 | -0.5 | 11.43 | 283.0608,268.0376,239.0358,163.0033,110.0009 | Wogonin | - | RS, L, RSLDP |
| 16 | $C_{16}H_{12}O_5$ | 283.0618 | 2.1 | 7.6 | 268.0373,239.0359, 195.0413,184.0544,163.0028 | Calycosin | 1.908 | L, RSLDP |
| 17 | $C_{16}H_{12}O_5$ | 283.0614 | 0.7 | 8.9 | 283.0631,268.0383,224.0476,198.0325,163.0042 | Prunetin | 2.99 | L, RSLDP |
| 18 | $C_{16}H_{12}O_6$ | 299.055 | -3.7 | 9.36 | 299.0500,284.0318,255.0491,153.9911 | 5,7,4'-Trihydroxy-8-methoxyflavone | - | RSLDP |
| 19 | $C_{16}H_{12}O_6$ | 299.0555 | -2 | 8.7 | 299.0585,284.0341,256.0380,227.0372,151.0030 | Kaempferide | 2.69 | RSLDP |
| 20 | $C_{16}H_{12}O_6$ | 299.056 | -0.4 | 11.46 | 299.0589,284.0321,256.0388,151.0048 | Chrysoeriol | 2.7496 | RSLDP |
| 21 | $C_{16}H_{14}O_4$ | 269.0818 | -0.5 | 11.8 | 269.0851,209.0614,183.0230,133.0289 | Retrochalcone | - | RSLDP |
| 22 | $C_{16}H_{14}O_5$ | 285.0761 | 2.7 | 7.7 | 285.0777, 191.0344,150.0333 | Licochalcone B | - | L |
| 23 | $C_{16}H_{14}O_6$ | 301.0707 | -3.6 | 7.31 | 269.0411,161.0211,139.0388,133.0235,124.0149 | 4',5,7-Trihydroxy-6-methoxyflavanone | - | RS, RSLDP |
| 24 | $C_{17}H_{14}O_6$ | 313.0721 | 1.2 | 11.7 | 283.0242,254.8550,211.0397,183.0440,155.0497 | 5,8-Dihydroxy-6,7-dimethoxyflavone | - | RS, RSLDP |
| 25 | $C_{17}H_{14}O_7$ | 329.0664 | -0.9 | 9.3 | 314.0492,229.1395,211.1360,171.1041,109.9991 | 5,7,2'-Trihydroxy-8,6'-dimethoxyflavone | - | RSLDP |
| 26 | $C_{17}H_{14}O_8$ | 345.06 | -4.7 | 7.55 | 345.0758,330.0404,315.0179,287.0261,149.0231 | Viscidulin III | - | RSLDP |
| 27 | $C_{19}H_{18}O_8$ | 373.0926 | -0.8 | 11.65 | 343.0473,328.0234,285.0042,257.0100,194.9926 | Skullcapflavone II | - | RS, RSLDP |
| 28 | $C_{20}H_{16}O_6$ | 351.087 | -1.2 | 11.85 | 283.1036,265.0850,241.0877 | Licoisoflavone B | - | RSLDP |
| 29 | $C_{20}H_{18}O_5$ | 337.1094 | 3.7 | 16.67 | 337.1102,293.1234,268.0370 | Eurycarpin A | - | RSLDP |
| 30 | $C_{20}H_{18}O_6$ | 353.1037 | 1.7 | 13.6 | 309.0439,297.0422,284.0350,269.0398,219.0630 | Licoisoflavone A | - | RSLDP |
| 31 | $C_{21}H_{18}O_{11}$ | 445.0768 | -1.9 | 7.3 | 445.0767,269.0460,241.0508,175.0234,113.0245 | Baicalin | - | RS, RSLDP |
| 32 | $C_{21}H_{18}O_{11}$ | 445.0767 | 0.1 | 7.57 | 269.0454,241.0511,197.0618,175.0228,113.0202 | Apigenin 7-O-glucuronide | - | RS, RSLDP |
| 33 | $C_{21}H_{20}O_{10}$ | 431.0967 | -3.9 | 7.31 | 269.0474,241.0580,223.0341 | Baicalein-7-O-β-D-glucoside | - | RS, RSLDP |
| 34 | $C_{21}H_{20}O_9$ | 415.1033 | 0.3 | 6.93 | 415.1043,295.0628,267.0651 | Chrysin -8-C-glucoside | 1.1664 | RS, RSLDP |
| 35 | $C_{21}H_{22}O_{10}$ | 433.1131 | -2.1 | 6.98 | 433.1140,271.0606, 177.0180,151.0039,119.0514 | Choerospondin | - | L, RSLDP |
| 36 | $C_{21}H_{22}O_{10}$ | 433.1126 | -3.2 | 6.42 | 433.1171,313.0605,271.0628,151.0046,119.0512 | Engeletin | - | RSLDP |

*(Continued)*

**Table 1.** (Continued)

| No. | Formula | [M −H]⁻ m/z | Error (ppm) | T$_R$ (min) | MS/MS fragments | Identification | Clog P | Source |
|---|---|---|---|---|---|---|---|---|
| 37 | $C_{21}H_{22}O_9$ | 417.1192 | 0.2 | 6.52 | 417.1196,255.0655,135.0099,119.0488,91.0194 | Liquiritin | - | L, RSLDP |
| 38 | $C_{21}H_{22}O_9$ | 417.1182 | -2.4 | 7.32 | 417.1195,255.0668,135.0088,119.0505,91.0155 | Isoliquiritin | - | L, RSLDP |
| 39 | $C_{22}H_{20}O_{11}$ | 459.0923 | -2.1 | 7.99 | 459.0978,283.0638,268.0401,163.0050,113.0258 | Wogonoside | 1.21 | RS, L, RSLDP |
| 40 | $C_{22}H_{20}O_{11}$ | 459.092 | -2.8 | 7.73 | 459.0952,283.0611,268.0379,175.0244,113.0245 | Oroxylin A-7-O-glucuronide | 0.963 | RS, RSLDP |
| 41 | $C_{22}H_{20}O_{12}$ | 475.0873 | -1.9 | 7.83 | 475.0742,299.0579,284.0341,253.0466,113.0232 | 5,7,2'-Trihydroxy-6-methoxyflavone-7-O-glucuronide | -0.0008 | RS, RSLDP |
| 42 | $C_{22}H_{20}O_{12}$ | 475.087 | -2.5 | 7.64 | 475.0828,299.0595,284.0289,253.0655,113.0232 | 5,7,8-Trihydroxy-6-methoxyflavone-7-O-β-D-glucuronide | 0.2491 | RS, RSLDP |
| 43 | $C_{22}H_{20}O_{12}$ | 475.0899 | 3.6 | 7.06 | 475.0915,299.0587,284.0341,175.0224,113.0236 | 5,7,2'-Trihydroxy-8-methoxyflavone-7-O-glucuronide | 1.44 | RS |
| 44 | $C_{22}H_{22}O_{10}$ | 445.1131 | -2 | 7.5 | 445.1157,283.0624,268.0389 224.0483 | Calycosin-7-O-glucoside | - | L, RSLDP |
| 45 | $C_{23}H_{22}O_{12}$ | 489.103 | -1.7 | 8.1 | 298.0410,283.0231,211.0396,175.0216,113.0253 | 5,7-Dihydroxy-2',8-dimethoxyFlavone7-O-glucuronide | - | RS, RSLDP |
| 46 | $C_{23}H_{22}O_{13}$ | 505.0984 | -0.8 | 7.74 | 459.0953 | Viscidulin II-2'-O-glucuronide | - | RSLDP |
| 47 | $C_{26}H_{28}O_{13}$ | 547.145 | -1.4 | 6.23 | 487.1257,457.1146, 427.1011,367.0810,337.0715 | Chrysin 6-C-arabinoside-8-C-glucoside | - | RS |
| 48 | $C_{26}H_{28}O_{14}$ | 563.1419 | 2.3 | 6.05 | 563.1410,473.1089,443.0961,383.0774,353.0687 | Schaftoside | - | RS, RSLDP |
| 49 | $C_{26}H_{28}O_{14}$ | 563.1379 | -4.8 | 6.06 | 563.1417,473.1071,443.0913,383.0780,353.0681 | Apigenin-7-apioglucoside | - | L |
| 50 | $C_{26}H_{30}O_{13}$ | 549.1606 | -1.4 | 6.4 | 549.1643,417.1219,255.0668,135.0097 | Liquiritin apioside | -0.2673 | L, RSLDP |
| 51 | $C_{26}H_{30}O_{13}$ | 549.1618 | 0.8 | 7.1 | 549.1643,417.1213,255.0677,135.0094 | Isoliquiritin apioside | -0.0154 | L, RSLDP |
| 52 | $C_{28}H_{30}O_{16}$ | 621.1436 | -4 | 6.8 | 621.1516,445.1147,430.0907,283.0590,268.0391 | Wogonin- O-glu-gluA | - | RS |
| 53 | $C_{29}H_{36}O_{15}$ | 623.1962 | -3 | 6.42 | 623.2023,547.1516,461.1657,415.0988,161.0252 | Acteoside | - | RS |
| 54 | $C_{30}H_{46}O_4$ | 469.3326 | 0.5 | 18.11 | 469.3320,425.3448,355.2637 | Glycyrrhetinic acid | - | L, RSLDP |
| 55 | $C_{30}H_{46}O_5$ | 485.3255 | -3.6 | 14.14 | 439.2838,425.3101,423.3114,411.3395,407.2924 | 24-Hydroxyglycyrrhetic acid | - | L |
| 56 | $C_{42}H_{62}O_{16}$ | 821.3965 | 0 | 9.82 | 821.4038,351.0572,193.0351 | Glycyrrhizic acid | - | L, RSLDP |
| 57 | $C_{42}H_{62}O_{16}$ | 821.3977 | 1.5 | 10.36 | 821.4033,645.3780,351.0570,193.0381,175.0239 | Licorice-saponin H2 | - | L, RSLDP |
| 58 | $C_{42}H_{62}O_{17}$ | 837.3926 | 1.4 | 8.35 | 837.4003,485.3247,351.0605,193.0350 | Licorice-saponin G2 | 1.73 | L, RSLDP |
| 59 | $C_{42}H_{62}O_{17}$ | 837.3917 | 0.3 | 9.17 | 837.4010,485.3100,351.0586 | hydroxy-Glycyrrhizic acid | - | L, RSLDP |
| 60 | $C_{42}H_{62}O_{17}$ | 837.3914 | -0.2 | 9.59 | 837.4004,485.3233,351.0569,193.0389 | Isomer of Licorice-saponin G2 | - | RSLDP |
| 61 | $C_{42}H_{64}O_{15}$ | 807.4193 | 2.5 | 11.91 | 807.4283,351.0602,193.0375 | Licorice-saponin B2 | - | RSLDP |
| 62 | $C_{42}H_{64}O_{16}$ | 823.4131 | 1 | 10.3 | 823.4195,351.0582,193.0360 | Licorice-saponin J2 | - | L, RSLDP |

flavone C-glycosides. Compound 47, as an example, gave [M-H]⁻ ions at m/z 547.1450, which fragmented into m/z 487.1257, 457.1146, 427.1011, 367.0810, and 337.0715 by losing of 60 Da, 90 Da, 120 Da, 180 Da, 210 Da from the precursor [M-H]⁻ ions. By comparing with literature data, the compound 47 was identified chrysin 6-C-arabinoside-8-C-glucoside [22].

*3.1.1.2 Flavone O-glycosides.* A total of 12 flavone O-glycosides were identified (peaks 31, 32, 33, 39, 40, 41, 42, 43, 45, 46, 49, 52). Generally, flavone O-glycosides gave [M-H-176]⁻ ions since the loss of glucuronic acid. Besides, RDA cleavages and the losses of $CH_3$ groups are the diagnostic fragmentation behaviors of flavone O-glycosides. For instance, compounds 39 and 40 both had [M-H]⁻ ions at m/z 459.0923, showing a molecular formula of $C_{22}H_{20}O_{11}$.Their MS/MS fragments gave a product ion at m/z 283.0611 with a loss of 176 Da. Their fragment ions at 268.0401, 163.0050 and 113.0258 yield by losses of $CH_3$, CHO, and RDA cleavage. Compared with reference standards, compound 39 was identified as Wogonoside, while compound 40 was identified as Oroxylin A-7-O-glucuronide.

**Table 2. Characterization of the metabolites in dosed plasma by UHPLC-QTOF-MS.**

| No. | Formula | [M –H]- $m/z$ | Error (ppm) | TR (min) | MS/MS fragments | Composition shift | Clog P | Source |
|---|---|---|---|---|---|---|---|---|
| M1 | $C_{22}H_{20}O_{11}$ | 459.0978 | -2.1 | 8.1 | 459.0978,283.0638,268.0401,163.0050,113.0258 | - | 1.2133 | RS, L, RSLDP |
| M2 | $C_{16}H_{12}O_5$ | 283.0608 | -0.5 | 11.56 | 283.0608,268.0376,239.0358,163.0033,110.0009 | - | 3.3306 | RS, L, RSLDP |
| M3 | $C_{15}H_{12}O_4$ | 255.0675 | -0.7 | 9.88 | 255.0675,213.0569,135.0095,119.0507,91.0200 | - | 2.7858 | RS, L, RSLDP |
| M4 | $C_6H_{12}O_6$ | 179.056 | -0.7 | 19.9 | 163.0128,151.0769,135.0458,71.0154,59.0156 | M1-$C_6H_{10}O_5$+Hydrogenation | -2.9027 | RS |
| M5 | $C_{15}H_{10}O_7S$ | 333.0075 | -4.8 | 9.15 | 333.0088,253.0504,209.0607 | M1-$C_6H_8O_6$ and $CH_2O$ +SulfateConjugation | 2.0028 | RS |
| M6 | $C_{15}H_{10}O_7S$ | 333.0075 | -4.3 | 6.87 | 253.0510,209.0591,135.0048 | M1-$C_6H_8O_6$ and $CH_2O$ +SulfateConjugation | 0.7758 | RSLDP |
| M7 | $C_{15}H_{10}O_8S$ | 349.0022 | -0.5 | 9.22 | 269.0448,223.0389,197.0616 | M1-$C_6H_8O_6$ and $CH_2$+SulfateConjugation | 1.4346 | RS, L, RSLDP |
| M8 | $C_{15}H_{10}O_8S$ | 349.0013 | -1 | 8.79 | 269.0459,223.0399,197.0603,155.0501 | M1-$C_6H_8O_6$ and$CH_2$+SulfateConjugation | 0.6046 | RS, RSLDP |
| M9 | $C_{15}H_{10}O_8S$ | 349.0027 | -1 | 8.49 | 269.0465,197.0609,171.0447 | M1-$C_6H_8O_6$ and$CH_2$+SulfateConjugation | 0.1445 | RSLDP |
| M10 | $C_{15}H_{12}O_7S$ | 335.0231 | -0.5 | 7.08 | 335.0243,255.0680,135.0089,119.0510,91.0233 | M3+SulfateConjugation | 1.2169 | L, RSLDP |
| M11 | $C_{15}H_{12}O_7S$ | 335.0215 | -4.9 | 8.17 | 335.0191,255.0668,135.0099,119.0477,91.0235 | M3+SulfateConjugation | 1.2258 | RSLDP |
| M12 | $C_{16}H_{12}O_3$ | 251.0723 | ,3.7 | 14.07 | 251.0685,233.1535,209.1554,194.0938,152.0841 | M1-$C_6H_8O_6$ and 2O | 3.6259 | RSLDP |
| M13 | $C_{16}H_{12}O_6$ | 299.0561 | ,0.6 | 9.31 | 299.0599,284.0333,125.9920 | M1-$C_6H_8O_6$+Oxidation | 2.7633 | RS |
| M14 | $C_{16}H_{12}O_7S$ | 347.0231 | -2.7 | 8.4 | 347.0202,267.0668,252.0456,223.0342,165.0317 | M1-$C_6H_8O_6$ and O+SulfateConjugation | 1.3618 | RSLDP |
| M15 | $C_{16}H_{12}O_7S$ | 347.0231 | -2 | 8.36 | 347.0213,267.0664,252.0426,223.0390 | M2-O+SulfateConjugation | 1.4753 | L |
| M16 | $C_{16}H_{12}O_8S$ | 363.018 | -0.4 | 9.17 | 363.0174,283.0607,268.0383,163.0039 | M2+SulfateConjugation | 1.9786 | RS, L, RSLDP |
| M17 | $C_{16}H_{12}O_8S$ | 363.0167 | -3.7 | 8 | 363.0169,283.0618,268.0363,163.0055 | M2+SulfateConjugation | 0.5436 | RS |
| M18 | $C_{21}H_{18}O_{11}$ | 445.0768 | -3.7 | 7.29 | 445.0767,269.0449,251.0341,175.0234 | M1-$CH_2$ | 0.7693 | RS |
| M19 | $C_{21}H_{20}O_{10}$ | 431.0984 | -3.9 | 7.31 | 431.1024,399.9724,269.0474 | M3+Glucuronidation | 0.8016 | RS |
| M20 | $C_{21}H_{20}O_9$ | 415.1042 | ,1.8 | 6.9 | 415.1043,325.0712,295.0628,267.0651 | M3-O+Glucuronidation | 0.6156 | RS, L |
| M21 | $C_{21}H_{22}O_9$ | 417.1191 | -2.3 | 6.48 | 417.1190,255.0664,135.0086,119.0493,91.0199 | M3+Glucuronidation | 1.2805 | RS |
| M22 | $C_{22}H_{20}O_{12}$ | 475.0882 | -1.9 | 7.83 | 475.0742,429.1935,299.0579,284.0341,253.0466 | M1+Oxidation | 0.5492 | RSLDP |
| M23 | $C_{22}H_{22}O_{10}$ | 445.1127 | -2.9 | 7.53 | 445.0934,430.0929,269.0463,197.0598,113.0250 | M2+GlucoseConjugation | 1.6922 | L |

*3.1.1.3 Free flavones.* In total, 12 compounds (5, 7–10, 15, 18, 20, 24–27) were considered as free flavones. Free flavones usually lose $H_2O$, CO, $CO_2$, methyl radical (methoxy group), and often undergo RDA cleavage to from fragmented ions. For example, compounds 7–10 all had a formula of $C_{15}H_{10}O_5$. Compound 7 was identified as baicalein by comparing with the reference standard. Compound 8 yield [M-H]- ions at m/z 269.0452 and fragment ions at 251.0368, 241.0526, 225.0543, 197.0608, and 171.0436 produced by losses of $H_2O$, CO, $CO_2$, and $C_2H_2O_3$. By comparing with previous reports, compound 8 was identified as Norwogonin [23]. Compounds 9 and 10 were identified as Apigenin and 5,7,2'-trihydroxyflavone by comparing the literature data [23,24].

*3.1.1.4 Flavanones, flavonol and flavanonol.* A total of 6 flavanones (peaks 12, 13, 23, 35, 37, and 50), 1 flavonol (peak 19), and 1 flavanonol (peak 36) were identified. RDA cleavages and sequential elimination of saccharides combined with neutral losses were found in the MS/MS spectra of flavanones, flavonol, and flavanonol. For instance, compound 50 showed an [M-H]- ions at m/z 549.1606. Its fragment ions at m/z 417.1219, 255.0668, and 135.0097 are due to the losses of apiosyl, glucosyl, and RDA cleavages. Thus, compound 50 was identified as Liquiritin apioside by the in-house library.

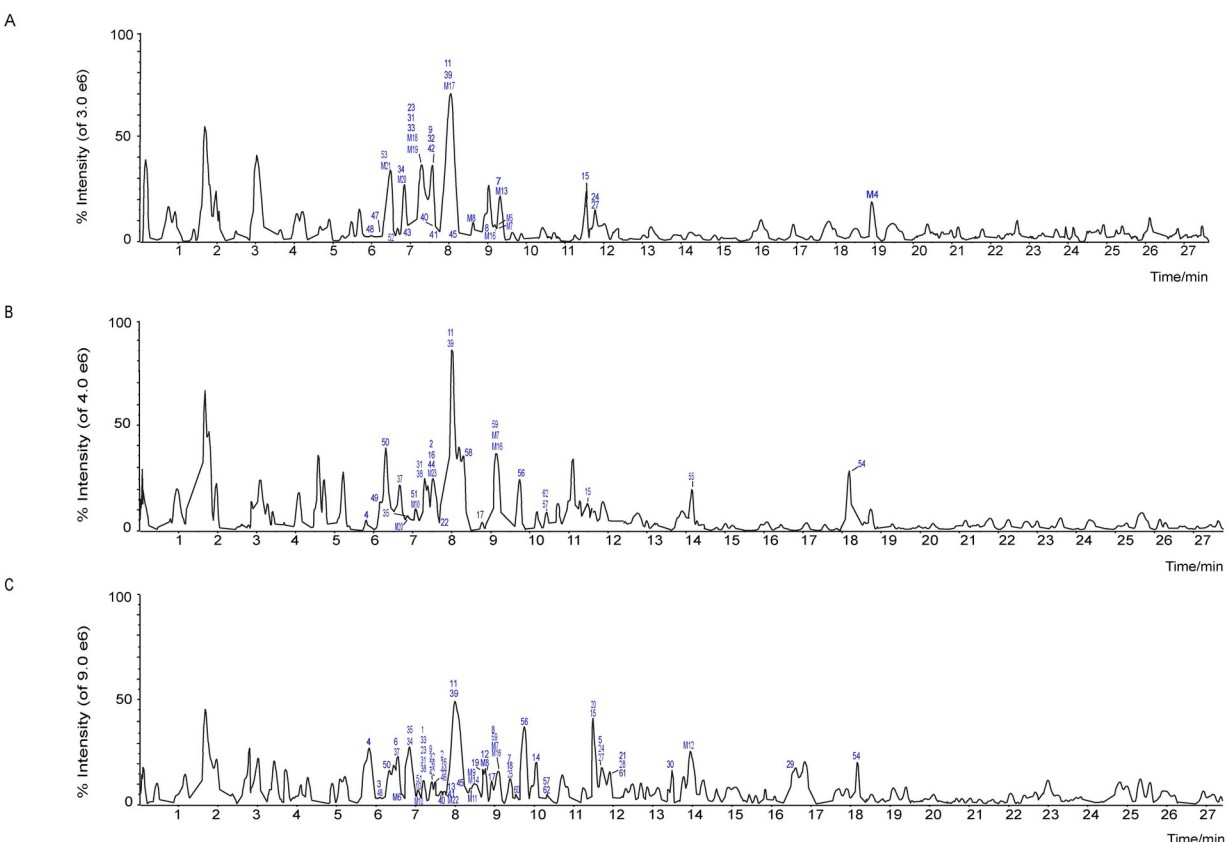

**Fig 1. Total ion chromatogram (TIC) of prototype components and metabolites in negative ion modes: (A): rat plasma after oral administration of Radix Scutellariae; (B): rat plasma after oral administration of Licorice; (C): rat plasma after oral administration of Radix Scutellariae-Licorice herb pair.**

*3.1.1.5 Chalcones and isoflavones.* Five chalcones (peaks 11, 21, 22, 38, 51) and eight isoflavones (peaks 6, 14, 16, 17, 28–30, 44) were identified. RDA cleavages and neutral losses are the typical characteristics of Chalcones. Take compound 11 as an example, it gave an [M-H]⁻ ions at m/z 255.0661, suggesting the formula $C_{15}H_{12}O_4$. Ions at m/z 135.0095, 119.0507 were attributed to result from RDA cleavages. By comparing the reference standards, compound 11 was identified as Isoliquiritigenin.

For isoflavones, we take compound 14 as an example. Compound 14 produced [M-H]⁻ ions at m/z 267.0662, and fragment ions at 252.0381, 223.0437, 195.0466, and 135.0080 yield by losses of a methyl radical, CHO, CO, and RDA cleavages. Compared with literature data, compound 14 was identified as Formononetin [25].

**3.1.2. Identification of triterpenoid saponins related prototype compounds.** A total of 9 triterpenoid saponins were identified in the prototype compounds. Generally, triterpenoid saponins yield an m/z 351 fragment ion due to the loss of glucuroglucuronic acid. For instance, compound 56 gave [M-H]⁻ ions at m/z 821.3965, and its fragment ions at m/z 351.0572, 193.0351 due to the loss of 2 glucuronic acid and $C_6H_6O_5$. Compared with the reference standard, compound 56 was identified as Glycyrrhizic acid.

## 3.2. Identification of metabolites of typical constituents in rat plasma

In this work, we selected Wogonin, Wogonoside, and Isoliquiritigenin as examples to elucidate how the metabolites produced from the absorbed constituents. A total of 23 metabolites of

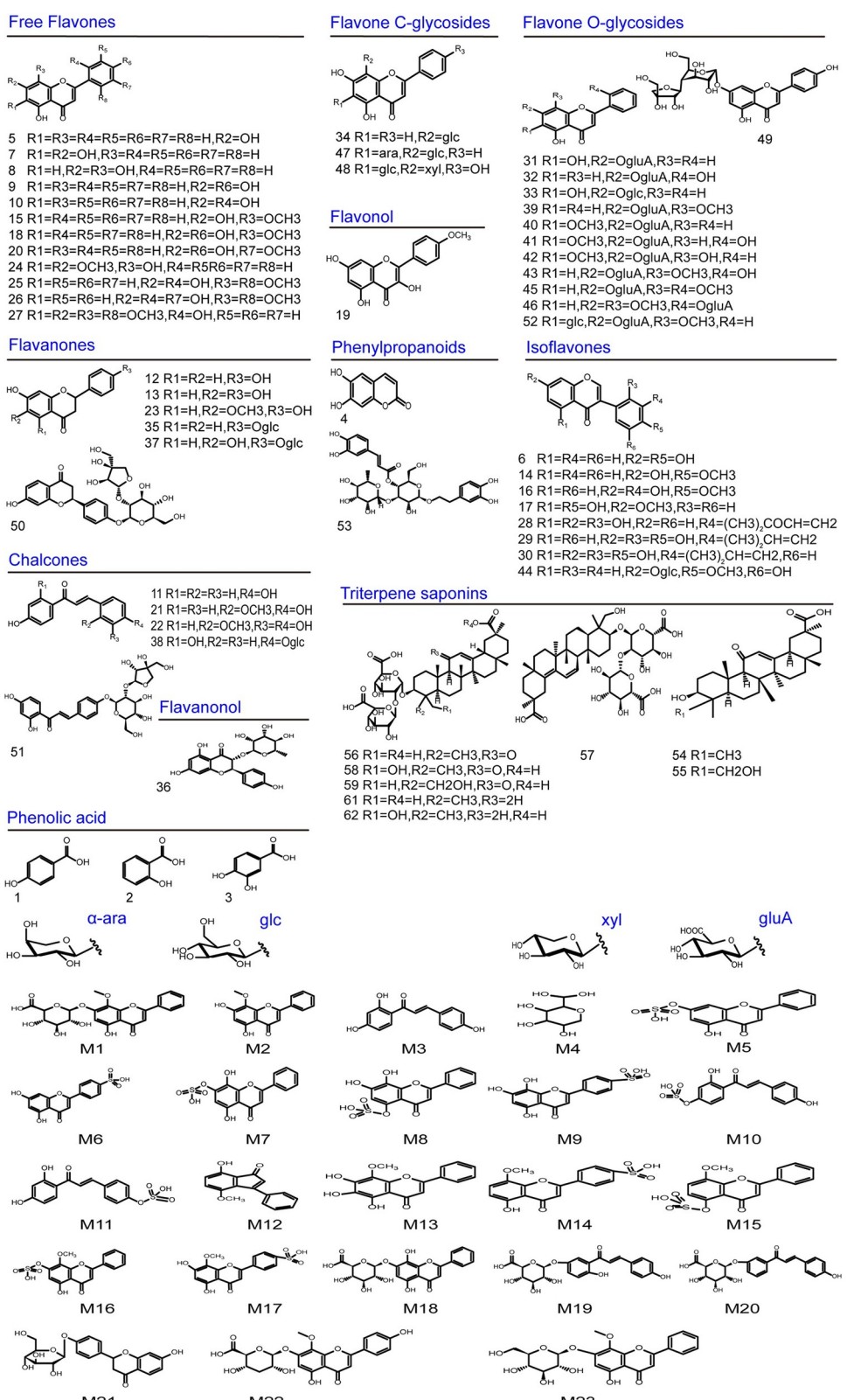

**Fig 2. Chemical structures of the absorbed prototypes and metabolites of Radix Scutellariae-Licorice herb pair.**

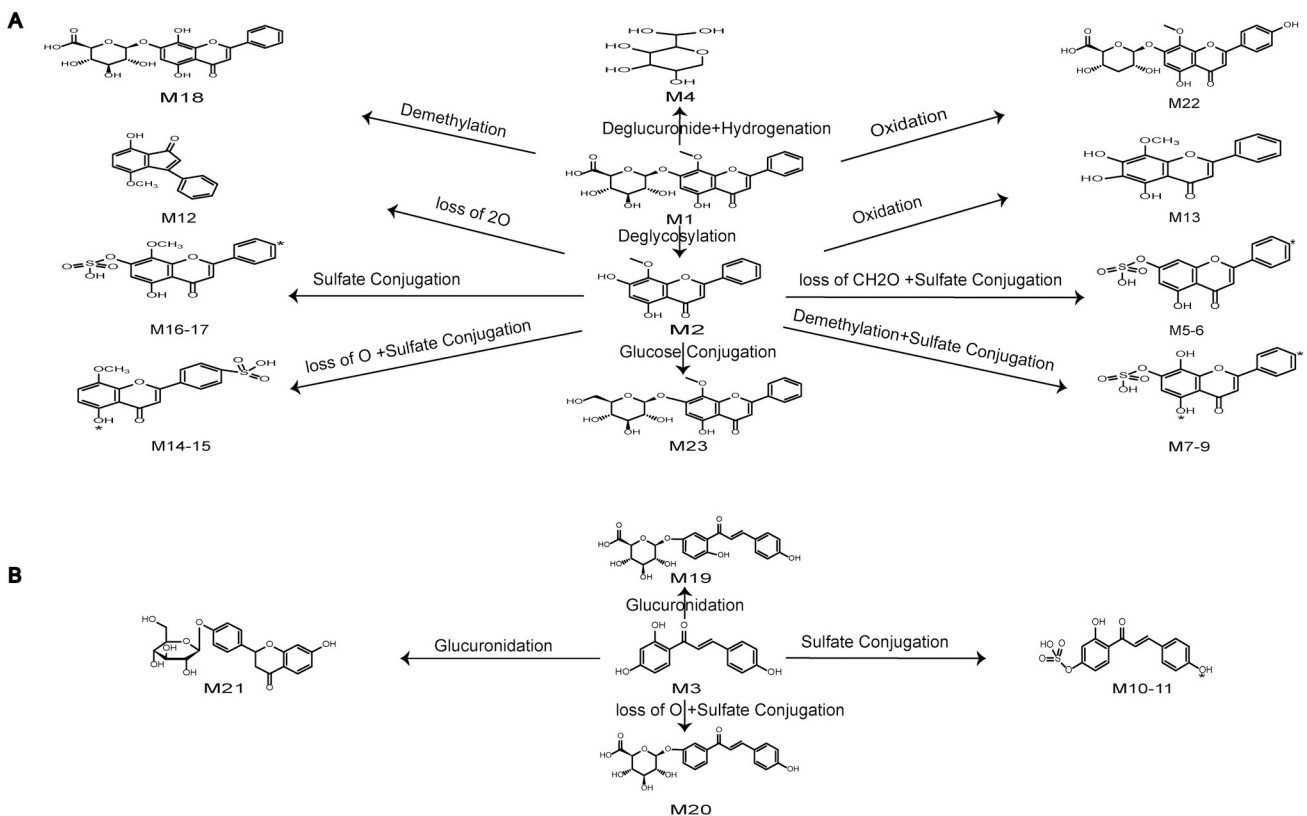

**Fig 3. Metabolic profile and proposed metabolic pathway of (A) Wogonoside, Wogonin and (B) Isoliquiritigenin.**

flavonoids were identified, which were produced by Wogonin, Wogonoside, and Isoliquiritigenin. The metabolic pathways of flavonoids are all shown in Fig 3. Oxidation, glucuronidation, diglucuronide, and deglycosylation are the main metabolic pathways of flavonoids. M1, M2, and M3 were identified as Wogonoside, Wogonin, and Isoliquiritigenin by comparing their retention time, accurate MS at m/z 459.0978, 283.0608, and 255.0675 [M-H]$^-$ and fragment ions with the reference standard, respectively. The possible fragmentation behavior and pathway of Wogonosid, Wogonin, and Isoliquiritigenin are shown in Figs 4–6.

**3.2.1. Identification of Wogonoside and Wogonin metabolites.** M4 provided a deprotonated molecular [M-H]$^-$ at m/z 179.0560 with a retention time of 19.9 min, 280 Da lower than of M1, which suggested that hydrogenation occurred on M1 after the loss of $C_6H_{10}O_5$. The major fragment ions identified at m/z 163.0128, 151.0769, and 135.0458 were yielded by the loss of oxygen atoms and carbon atoms. The formula of M4 was identified as $C_6H_{12}O_6$. The deprotonated molecular [M-H]$^-$ of M18 was detected at m/z 445.07677, which was 14 Da less than M1. The product ions at m/z 269.0449, 251.0341, and 175.0234 suggested that it might be demethylation. The formula was identified as $C_{21}H_{18}O_{11}$.

M22 was characterized as deprotonated molecular ions [M-H]$^-$ at m/z 475.0882, which was 16 Da higher than the value obtained for M1. The finding indicated that M22 was oxidation metabolite of M1, and the formula was $C_{22}H_{20}O_{12}$. The diagnostic fragment ions at m/z 299.0579 and 284.0341 were yielded by the loss of $C_6H_8O_6$ and $CH_3$.

M5 and M6 were eluted at 9.15 and 6.87 min, respectively, with deprotonated molecular ions [M-H]$^-$ at m/z 333.0075, which was 50 Da higher than M2. The fragment ions at m/z 253.0504, 209.0607, and 135.0048 were generated by the loss of $SO_3$, $CO_2$, and $C_6H_2$, which

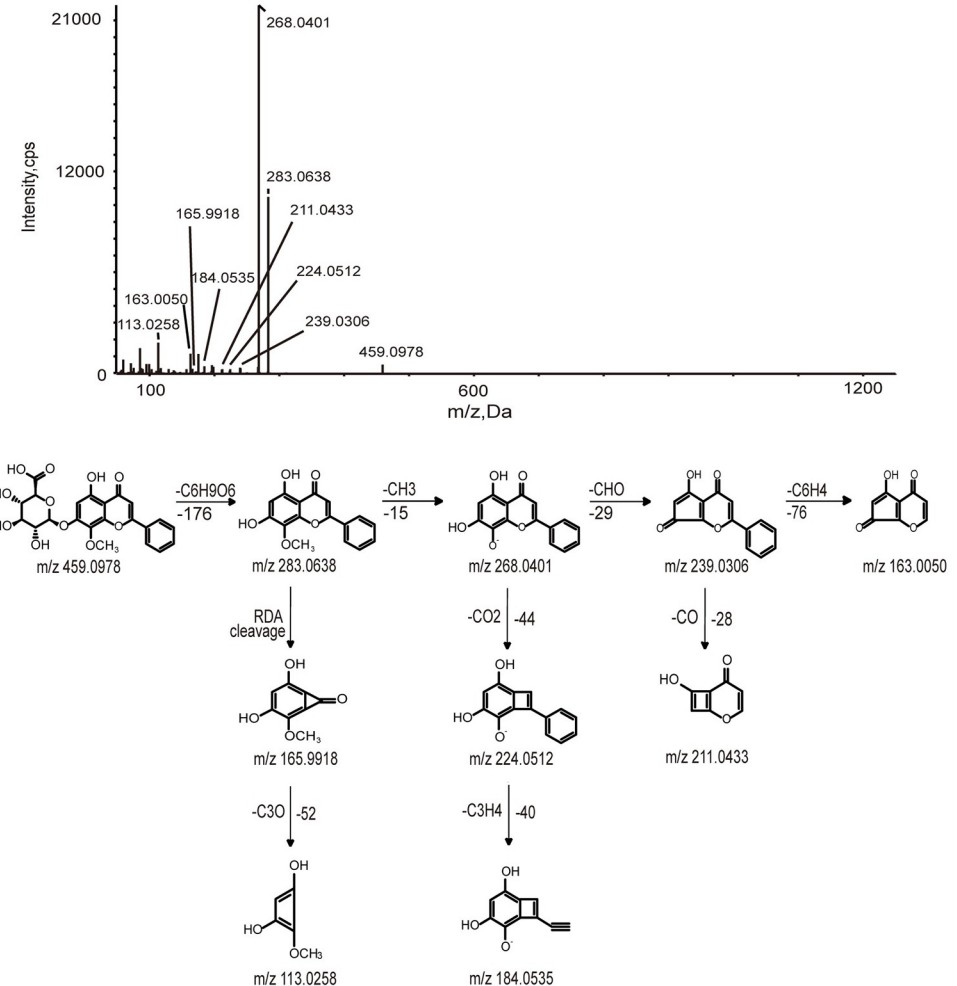

**Fig 4. MS/MS spectrum and cleavage pathway of Wogonoside.**

suggested that the formula was $C_{15}H_{10}O_7S$. Then, M5 and M6 were identified based on their clog P values of 2.0028 and 0.77575, respectively.

M7-M9 were identified as sulfate conjugation metabolites. M7-M9 were eluted at 9.22, 8.79, and 8.49 min and had deprotonated molecular ions $[M-H]^-$ at m/z 349.0022, 349.0013, and 349.0027, which were 66 Da higher than the value obtained for M2. The product ions identified at m/z 269.0448, 223.0389, and 195.0453 were produced by the loss of $SO_3$, CO, and O, which suggested that the formula was $C_{15}H_{10}O_8S$. In addition, clog P values of M7-M9 were 1.4346, 0.6046, and 0.1445.

M12 expressed $[M-H]^-$ at m/z 251.0723, which was yielded by the loss of 2 oxygens from M2. The typical fragment ions at m/z 233.1535, 209.1554, and 194.0938 were detected by loss of $H_2O$, 2C, and CH3, which suggested that the formula was $C_{16}H_{12}O_3$.

M13 showed $[M-H]^-$ at m/z 299.05611, 16 Da higher than M2, which implied that M13 was an oxidation metabolite. The fragment ions at m/z 284.0333 and 125.9920 were produced by the loss of CH3, $C_2HO_2$, CO, and $C_6H_2$. Thus, the chemical formula of M13 was $C_{16}H_{12}O_6$.

M14 and M15 exhibited the $[M-H]^-$ at m/z 347.0231, and had retention times of 8.40 and 8.36 min, respectively. The masses of the deprotonated M14 and M15 were 64 Da higher than the mass of M2, which indicated that sulfate conjugation occurred on M2 after the loss of O.

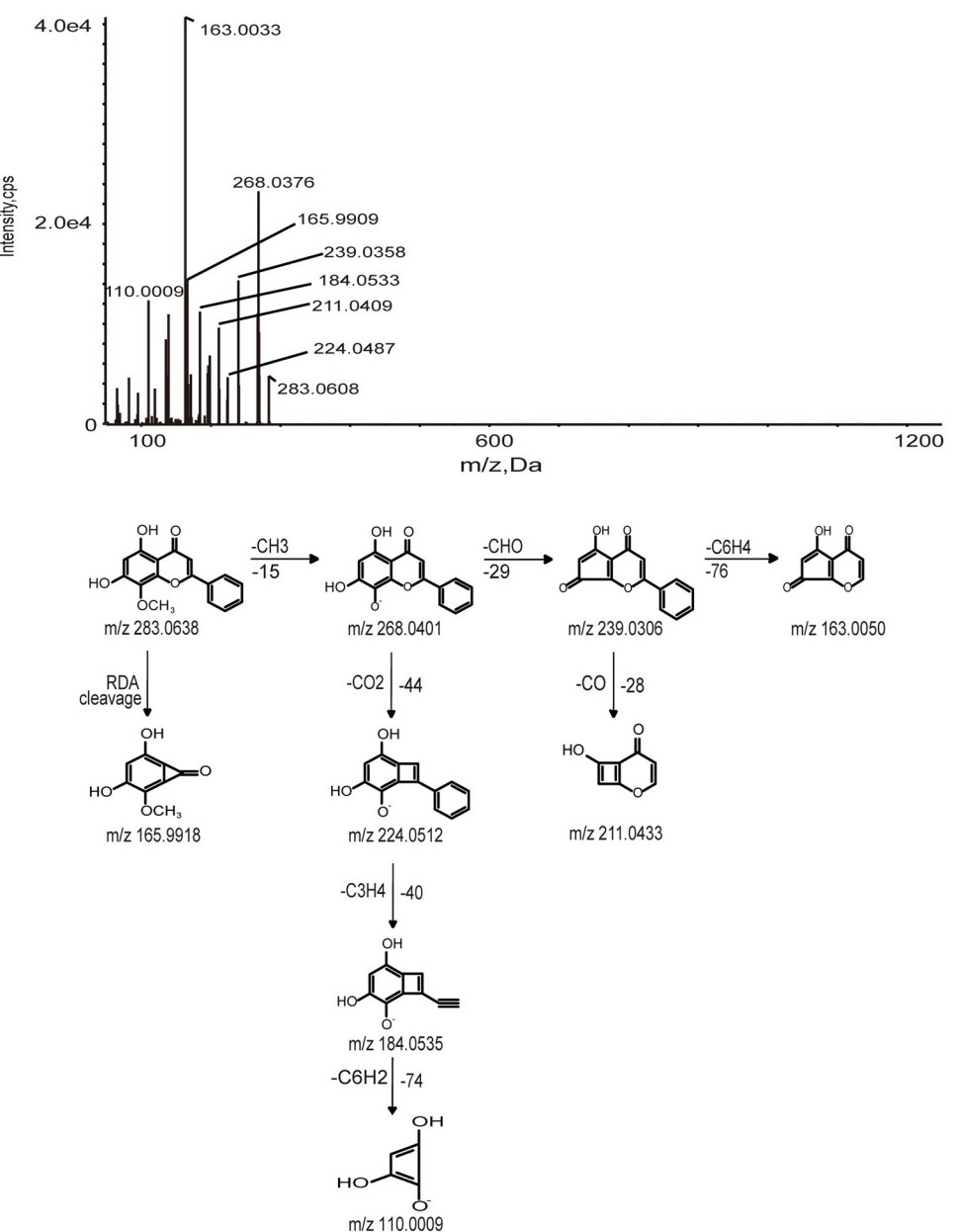

**Fig 5. MS/MS spectrum and cleavage pathway of Wogonin.**

The fragment ions at m/z 267.0664, 252.0426, and 223.0390 were formed by the loss of $SO_3$, $CH_3$, and CHO, which suggested that the formula was $C_{16}H_{12}O_7S$. Moreover, M14 and M15 were confirmed based on clog P values of 1.3618 and 1.47525, respectively.

M16 and M17 were eluted at 9.17 and 8.0 min with deprotonated molecular ions [M-H]⁻ at m/z 363.0180 and 363.0167, respectively, which were 80 Da higher than M2. Secondary fragment ions were founded at m/z 283.0618 [M-SO₃-H]⁻, 268.0363 [M-SO₃-CH₃-H]⁻, and 163.0055 [M-SO₃-CH₃-C₆H₄-CHO-H]⁻, suggesting that M16-M17 had undergone sulfate conjugation, indicating that the molecular formula was $C_{16}H_{12}O_8S$. Besides, the clog P values of M16 and M17 were 1.9786 and 0.5436, respectively. Hence, M16 and M17 were confirmed based on the retention times and clog P values.

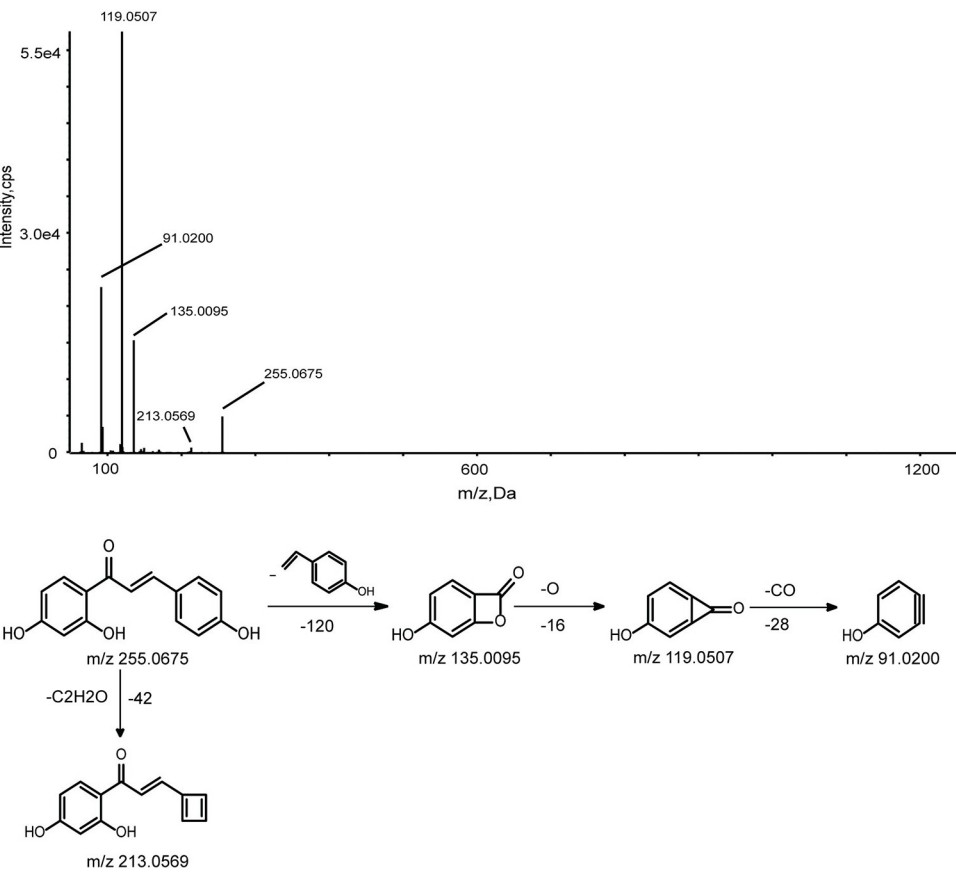

**Fig 6. MS/MS spectrum and cleavage pathway of Isoliquiritigenin.**

M23 produced a deprotonated molecular [M-H]$^-$ at m/z 445.1127, which was 162 Da higher than the value obtained for M2. The fragment ions at m/z 430.0929, 269.0463, and 197.0598 were produced by the loss of $CH_3$, $C_6H_{10}O_5$, $CH_2$, $CO_2$, and CO. According to the fragment ions, the chemical formula of M23 was $C_{22}H_{22}O_{10}$.

**3.2.2 Identification of Isoliquiritigenin metabolites.** M10-M11 were eluted at 7.08 and 8.17 min and had deprotonated molecular ions [M-H]$^-$ at m/z 335.0231 and 335.0215, which were 80 Da higher than the value obtained for M3. The major product ions formed at m/z 255.0680, 135.0089, and 119.0510 were produced by the loss of $SO_3$, $C_8H_7O$, O, and CO, which implied that the formula was $C_{15}H_{12}O_7S$. In addition, clog P values of M10-M11 were 1.2169 and 1.2258. Therefore, M10 and M11 were identified based on the retention times and clog P values.

M19 gave [M-H]$^-$ ions at m/z 431.09837, which was 176 Da higher than M3, suggesting that M19 was a glucuronide conjugation metabolite. The major product ions formed at m/z 399.9724 and 269.0474 were yielded by the loss of two O, $C_4H_6O$ and CO, which implied that the formula was $C_{21}H_{20}O_{10}$.

M20 gave [M-H]$^-$ ions at m/z 415.1042, which was 160 Da higher than M3, suggesting that sulfate conjugation occurred on M3 after the loss of O. The fragment ions formed at m/z 325.0712, 295.0628, and 267.0651 were generated by the loss of $C_3H_6O_3$, $CH_2O$, and CO, which suggested that the formula was $C_{21}H_{20}O_9$.

M21 exhibited a deprotonated ion [M-H]$^-$ at m/z 417.11911, which was 162 Da higher than the value obtained for M3. In addition, a series of typical fragment ions were detected at m/z

255.0664, 135.0086, and 119.0493 due to the loss of $C_6H_{10}O_5$ and RDA cleavages, implying that the chemical formula was $C_{21}H_{22}O_9$.

## 3.3 Compound targets of RSLDP

From the Swiss Target Prediction database, 440 targets were gathered corresponding to the selected 51 absorbed components (after removing ingredients without any relevant targets) and their 10 metabolites in RSLDP.

## 3.4 COVID-19 targets

1341 COVID-19-related differential genes from GeneCards, OMIM, PharmGkb, TTD, and Drugbank databases were collected.

## 3.5 PPIs

A total of 106 overlapping targets were identified by intersecting of the drug-related targets with disease-related targets. In detail, 10 genes were only in Radix Scutellariae-COVID-19 co-targets, 32 genes were only in Licorice-COVID-19 co-targets, and 64 genes were both in Radix Scutellariae-COVID-19 co-targets and Licorice-COVID-19 co-targets (Fig 7A). To obtain the potential relationship among the 106 targets of RSLDP against COVID-19, the PPIs were imported into the STRING database. PPIs containing 100 nodes and 489 edges were acquired (Fig 8). In the figures shown here (Fig 8), every circle expressed a target protein, and the center of the dot represented the protein structure.

## 3.6 Network construction

To clarify the potential mechanism of RSLDP in the treatment of COVID-19, Cytoscape v3.9.1 software was adopted to construct a compounds-targets network, as shown in Fig 7B.

The red, teal circle stands for compounds of Radix Scutellariae and Licorice, respectively. The red and teal circle represents common compounds of Radix Scutellariae and Licorice. A blue rectangle is employed to stand for the target gene. And edges symbols targets interacting with them. As presented in Fig 7B, the network shows 167 nodes, containing 61 compounds, 106 target genes as well as 797 edges. The average degree value is 9.545, and the degree of 63 nodes was larger than 9.545. The results indicate that these nodes may be candidate active compounds and potential targets for RSLDP in the treatment of COVID-19.

We used Cytoscape v3.9.1 software to construct a PPIs network. (Fig 9) As shown in Fig 9, a network contained 100 nodes and 489 edges. The sub-network 1 contained 33 nodes and 243 edges. The sub-network 2 contained 12 nodes and 61 edges. Besides, we used a plugin, CytoNCA, to screen core targets according to BC, DC, CC and EC. The threshold values were BC >6.455, DC >13, CC >0.627, and EC >0.16. Based on DC, the core targets were as follows: STAT3, AKT1, EGFR, HSP9AA1, MAPK3, JUN, IL6, VEGFA, TNF, IL2, RELA, and STAT1.

## 3.7 Pathway enrichment analysis

To explore the functional distribution of RSLDP in the treatment of COVID-19, the 106 targets were sent to R software, and the enrichment analysis of GO and KEGG pathways was performed. As shown in Fig 10A, we selected 10 items of biological function, cell function, and molecular function in GO analysis, which involved protein kinase activity, response to chemical stress, response to oxygenates, positive regulation of cytokines, and other biological process functions. As shown in Fig 10B, it shows the first 20 signaling pathways, which relate to PI3K-Akt signaling pathway, AGE-RAGE signaling pathway for diabetic complications, virus-

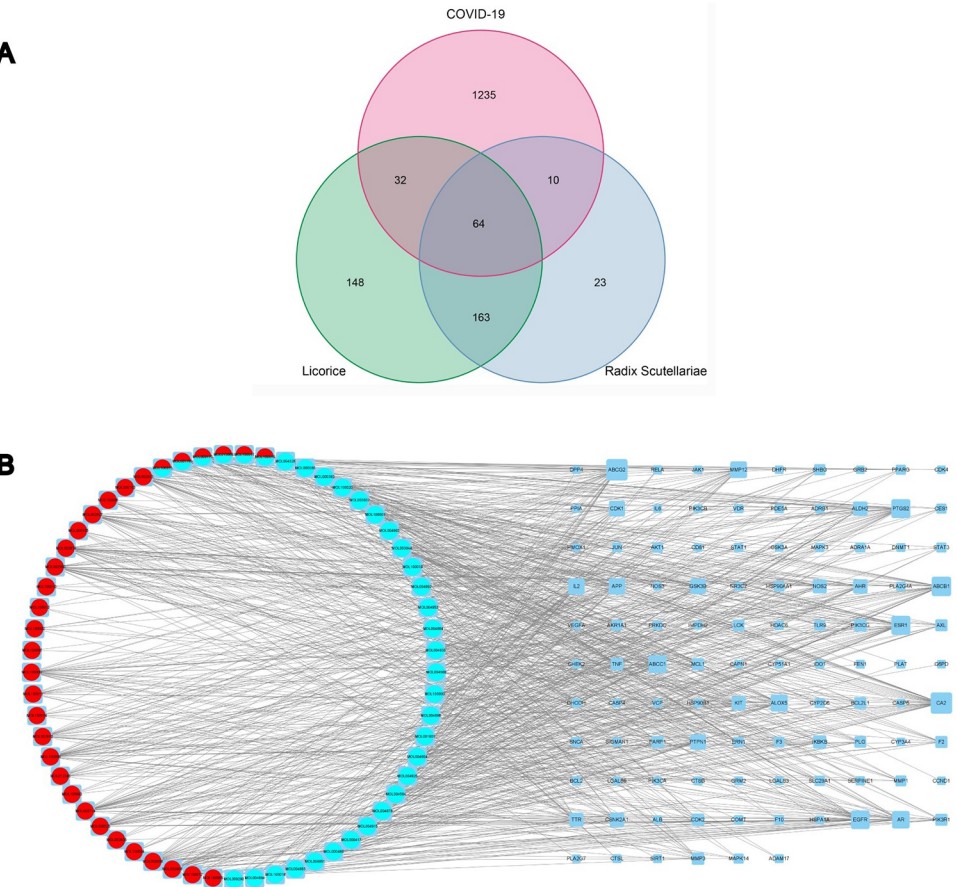

**Fig 7. Interactions between herbs,RSLDP-related targets and COVID-19-related targets.** (A): Overlaps of RSLDP potential targets and COVID-19 related targets are shown in a Venn diagram. (B): The interaction network of targets of the abosorbed compounds and the related target proteins.Red, teal circle stands for compounds of Radix Scutellariae and Licorice respectively. Red and teal circle represents common compounds of Radix Scutellariae and Licorice. Blue rectangle is employed to stand for target gene. And edges symbols targets interacting with them.

related pathways such as novel coronavirus and human cytomegalovirus infection, inflammatory immune-related pathways, and so on.

Traditional Chinese medicines (TCM) have the effect characteristics of multi-component, multi-pathway, and multi-target in treating diseases. For oral herbal drug pairs, it is commonly believed that their components must be absorbed into the blood to possess therapeutic effects. The holistic and systematic characteristics of target-network pharmacology correspond to the study content of "multi-component, multi-pathway, and multi-target" in TCM. The combination of blood components and target-network pharmacology is useful in identifying the absorbed ingredients of drug pairs and predicting their mechanism of action. Based on this approach, we combined the 61 blood components of RSLDP with its target-network pharmacology.

From the analysis results of prototype components of RSLDP before and after pairing, most of the components in the group of RSLDP can be found in the individually administered group, but 20 components were detected after the pairing that were not detected before the pairing. This may be the result of the interaction of the components of the herbs during the pairing process. It also indicates that the pairing promotes the selective absorption of these components. However, the research is just a preliminary study of the overall changes in the

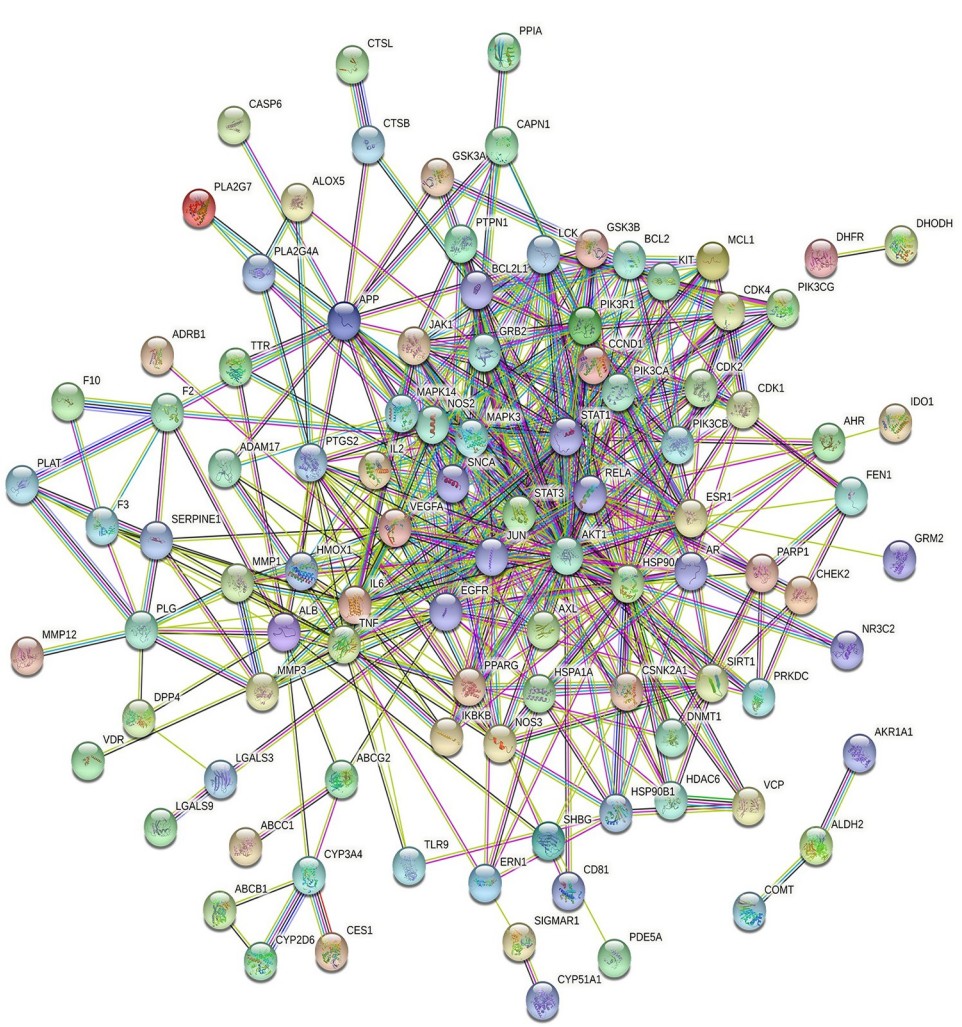

**Fig 8. Protein-protein interactions (PPIs) of Radix Scutellariae-Licorice herb pair.**

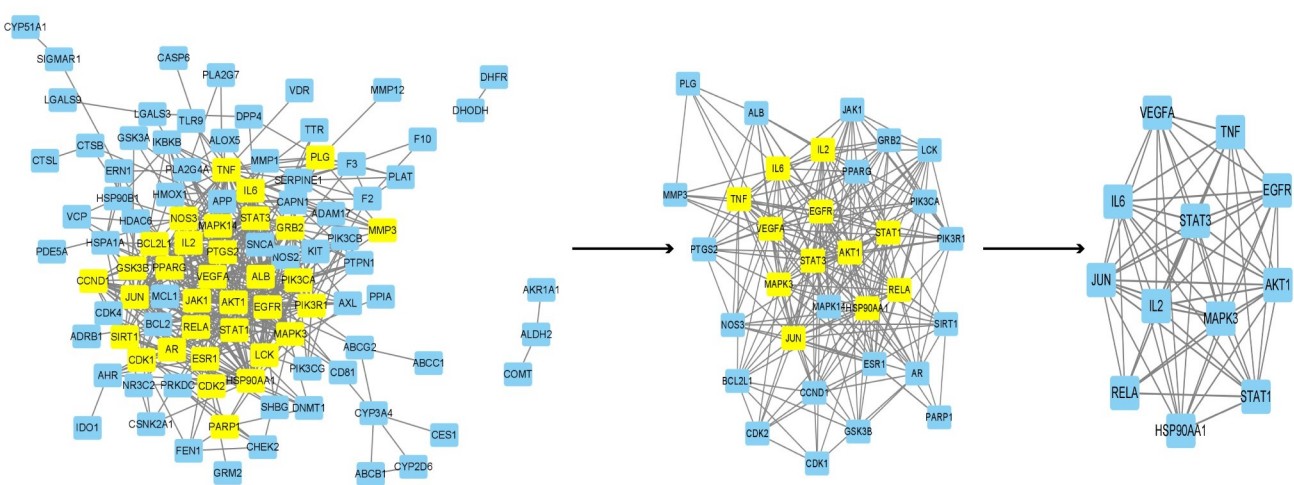

**Fig 9. The process of topological screening for the protein-protein interactions (PPIs) network.**

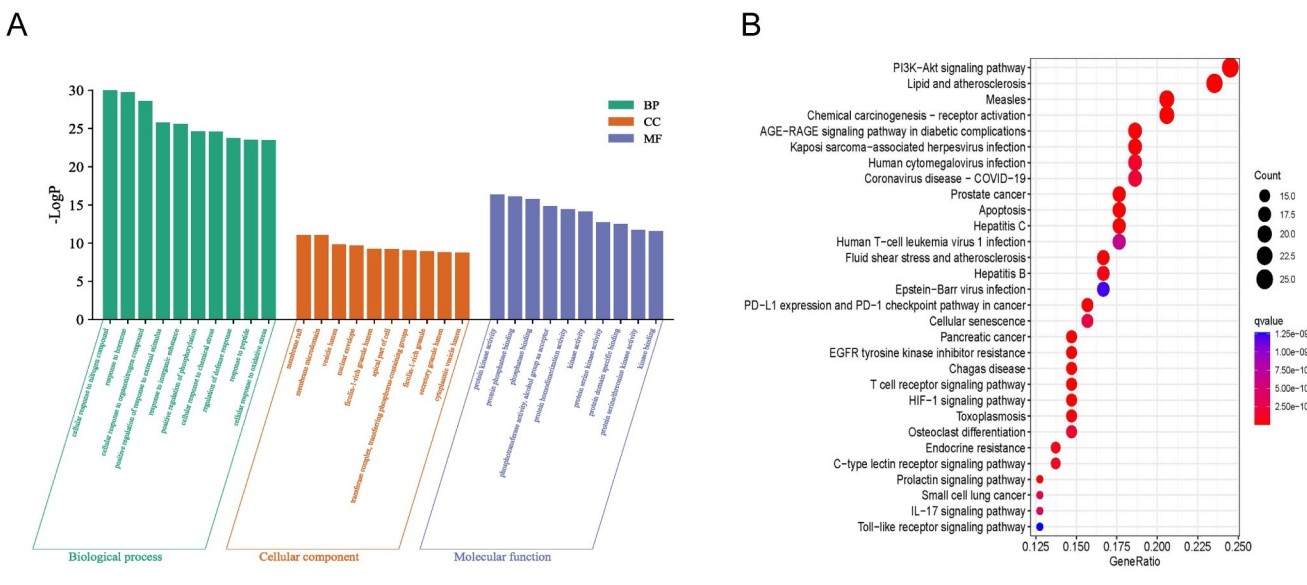

**Fig 10. GO and KEGG pathway analysis.** (A) GO analysis. (B) KEGG analysis.

chemical composition of the RSLDP in vivo before and after the pairing. Therefore, we will explore the differences in the component content and metabolites in vivo before and after the pairing to more comprehensively explain the chemical mechanism of the pairing in future research.

From the results of target-network pharmacology, 61 compounds and 106 targets may be the bioactive compounds and pharmacological targets of RSLDP for the treatment of COVID-19. Further research revealed that seven key active components (chrysin, baicalein, wogonin, apigenin, isoliquiritigenin, isoliquiritin apioside, and Glycyrrhetinic acid) were regarded to be effective on COVID-19, and twelve targets (MAPK3, IL6, JUN, IL2, STAT3, AKT1, TNF, STAT1, EGFR, HSP90AA1, VEGFA RELA) were screened to be effective on COVID-19. Studies have shown that baicalein and apigenin can inhibit the 3C-like protease of SARS-CoV-2 [26]. Glycyrrhetinic acid has been shown to have anti-COVID-19 activity [27]. Isoliquiritin apioside is also a potential compound against COVID-19 [28]. IL-6 has been shown to be associated with COVID-19 [29]. It has been shown that JUN and AKT1 play an important role in COVID-19 [30,31]. Increased levels of TNF, a key pro-inflammatory cytokine, have been proven to be related to elevated mortality in COVID-19 [32]. STAT3 is a potential molecular target for clinical syndromes characterized by systemic inflammation in COVID-19 [33]. KEGG pathway analysis further indicated that the most important of RSLDP were PI3K-Akt signaling pathway, AGE-RAGE signaling pathway for diabetic complications, and inflammatory immune-related pathways. This result indicated that RSLDP exerts its therapeutic effect by modulating the immune response and inflammatory activation processes. The PI3K-Akt signaling pathway is involved in the pathogenesis of pulmonary fibrosis and in the immune response process of host cells to resist viral invasion, and this pathway is significant in the anti-inflammatory effects of COVID-19 [34]. GO pathway analysis further indicated that the most important of RSLDP were protein kinase activity, response to chemical stress, response to oxygenates, positive regulation of cytokines, and other biological process functions. It has been shown that their responses may play a driving role in the development and progression of COVID-19, such as the cellular response to chemical stress, and response to oxygenated materials [35]. Though some results have been obtained from this study, there are still limitations.

The targets of components should be further verified, including existing compounds and functional proteins.

## 4. Conclusion

In the current study, we combined UHPLC-QTOF-MS analysis and target network pharmacology to expose the potential pharmacological mechanisms of RSLDP against COVID-19. The identified compounds by UHPLC-QTOF-MS analysis provided a material basis for target network pharmacology. The results have shown that the effects of 61 active compounds of RSLDP on the 106 potential targets were related to COVID-19. Based on it, the twelve core targets (STAT3, AKT1, EGFR, HSP9AA1, MAPK3, JUN, IL6, VEGFA, TNF, IL2, RELA, and STAT1) could be the most important targets for RSLDP in the treatment of COVID-19. And it may play a therapeutic role via PI3K-Akt signaling pathway, AGE-RAGE signaling pathway for diabetic complications, virus-related pathways such as novel coronavirus infection, inflammatory immune-related pathways, and so on. Overall, this is the first report to employ UHPLC-QTOF-MS analysis on the metabolic pathways of RSLDP in vivo. The findings not only clarify the potential mechanisms of RSLDP in the treatment of COVID-19 but also provide a theoretical basis for the future clinical research of TCMs.

## Supporting information

**S1 File. List of intersections targets and involved-compounds.**
(DOCX)

## Acknowledgments

Thanks for instrument support from the Department of Pharmaceutical Analysis, School of Pharmacy, Hebei Medical University.

## Author Contributions

**Data curation:** Xuqing Wen, Weiwei Xie, Juan Gao, Dedong Zhang, Mengxin Yang.

**Formal analysis:** Juan Gao.

**Methodology:** Xuqing Wen.

**Software:** Xuqing Wen.

**Writing – original draft:** Xuqing Wen.

**Writing – review & editing:** Weiwei Xie, Zhiqing Zhang, Yingfeng Du, Yiran Jin.

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
