## [Decision Letter · Decision Letter 0]

11 Apr 2023

PONE-D-23-05859Systematically uncovering the absorbed effective substances of Radix scutellaria-licorice drug pair in rat plasma against COVID-19 using a combined UHPLC-Q-TOF-MS analysis and target network pharmacologyPLOS ONE

Dear Dr. yiran,

Thank you for submitting your manuscript to PLOS ONE. After careful consideration, we feel that it has merit but does not fully meet PLOS ONE’s publication criteria as it currently stands. Therefore, we invite you to submit a revised version of the manuscript that addresses the points raised during the review process.

We look forward to receiving your revised manuscript.

Kind regards,

Chun-Hua Wang

Academic Editor

PLOS ONE

Journal Requirements:

"This study was financially supported by Hebei Administration of Traditional Chinese Medicine (No. 2021133), the Natural Science Foundation of Hebei Province of China (H2019206562) and the Key Projects of Hebei Education Department (No. ZD2017244)."

"Data curation: X.Q.Wen, J.Gao, D.D.Zhang; design of the study: X.Q.Wen, W.W.Xie; statistical analysis:X.Q.Wen, J.Gao; analysis and interpretation of the data:X.Q.Wen, M.X.Yang; drafting the manuscript: X.Q.Wen,W.W.Xie; critical revision of the manuscript: Y.R.Jin, Y.F.Du, Z.Q.Zhang."

Reviewers' comments:

Reviewer's Responses to Questions

**Comments to the Author**

1. Is the manuscript technically sound, and do the data support the conclusions?

Reviewer #1: Partly

Reviewer #2: Partly

2. Has the statistical analysis been performed appropriately and rigorously? 

Reviewer #1: N/A

Reviewer #2: N/A

3. Have the authors made all data underlying the findings in their manuscript fully available?

Reviewer #1: Yes

Reviewer #2: Yes

4. Is the manuscript presented in an intelligible fashion and written in standard English?

Reviewer #1: No

Reviewer #2: No

5. Review Comments to the Author

Reviewer #1: 1. The English is too poor.

2. In section 2.7.1, Please describe how to search targets by Swiss Target Prediction, using a SMILES or draw a molecule.

3. The network pharmacology should add the Gene Ontology and KEGG analysis.

Reviewer #2: The study “ Systematically uncovering the absorbed effective substances of Radix scutellaria-licorice drug pair in rat plasma against COVID-19 using a combined UHPLC-Q-TOF-MS analysis and target network pharmacology” provides valuable insights into the identification of RS-related, Licorice-related, and RSL-related compounds in rat plasma using mass spectrometry. The authors have identified a range of prototype compounds and metabolites in the different groups, which adds to the existing knowledge in the field.

However, the consistent writing mistakes, punctuation errors, and formatting inconsistencies throughout the paper reflect a lack of respect to the journal and reviewers, which is concerning.

1. The text contains grammatical errors throughout the manuscript. Punctuation and spacing are often missing, making it difficult to read. The reference formatting appears to be inconsistent. A thorough proofreading and editing are necessary.

• e.g. In abstract: "Besides, we optimized the conventional process ways of network pharmacology...", consider rephrasing it to "Additionally, we optimized the conventional methodologies of network pharmacology..." for better clarity.

• “A total of 85 absorbed constituents were identified or tentatively characterized in dosed plasma,including 32 components in Radix Scutellaria,27 components in Licorice and 65 components in RSLDP.” There is lack of space after each punctuation, and the sentence is unclear and hard to follow.

2. There is a lack of context and background information in the results and discussion section. It would be helpful for the authors to better explain the rationale behind the study, the significance of the findings, and how they relate to existing knowledge in the field.

3. The manuscript would benefit from a more detailed discussion of the limitations of the study, as well as suggestions for future research.

Section 3, which is titled “results and discussion”, does not adequately discuss the gap between the results and the literature. Only four pieces of literature are cited in this section, which is unusual for a discussion section.

4. The figures should be clear and of high quality.

e.g. Figure 9 has been stretched strangely. Some gene names are not fully displayed or difficult to read.

5. The format of references is seriously inconsistent. The references should be formatted consistently, and the authors should ensure that all cited literature is relevant and up-to-date.

6. PLOS authors have the option to publish the peer review history of their article (what does this mean?). If published, this will include your full peer review and any attached files.

Reviewer #1: No

Reviewer #2: No

---

## [Author Response · Author response to Decision Letter 0]

12 Jun 2023

Dear Editor,

Thank you for your letter and comments concerning our manuscript entitled '' Systematically uncovering the absorbed effective substances of Radix Scutellaria-licorice drug pair in rat plasma against COVID-19 using a combined UHPLC-Q-TOF-MS analysis and target network pharmacology''. (Manuscript Number: PONE-D-23-05859). We have carefully revised the manuscript in accordance with the reviewers’ comments and read the manuscript to minimize typographical, grammatical, and bibliographical errors.The revised portion is marked in red bold. We would like to present the alterations and explanations to your comments point-by-point as follows:Reviewer #1:

Comment 1: The English is too poor.

Reply: Thanks for your comments. We have tried our best to edit the language in the revised manuscript.

Comment 2: In section 2.7.1, Please describe how to search targets by Swiss Target Prediction, using a SMILES or draw a molecule.

Reply: Thanks for your comments. We have added more details on how to search for targets from the Swiss Target Prediction database in section 2.7.1.

Comment 3: The network pharmacology should add the Gene Ontology and KEGG analysis.

Reply: Thank you very much for your helpful suggestion. Based on your suggestions, the Gene Ontology and KEGG analysis have been added to the revised manuscript.

Reviewer #2:

Comment 1: The text contains grammatical errors throughout the manuscript. Punctuation and spacing are often missing, making it difficult to read. The reference formatting appears to be inconsistent. A thorough proofreading and editing are necessary.

• e.g. In abstract: "Besides, we optimized the conventional process ways of network pharmacology...", consider rephrasing it to "Additionally, we optimized the conventional methodologies of network pharmacology..." for better clarity.

• “A total of 85 absorbed constituents were identified or tentatively characterized in dosed plasma, including 32 components in Radix Scutellaria, 27 components in Licorice and 65 components in RSLDP.” There is lack of space after each punctuation, and the sentence is unclear and hard to follow.

Reply: Thank you very much for your helpful suggestion. We were really sorry for our careless mistakes. We have revised the grammar, spelling and reference formatting of the entire manuscript.

"Besides, we optimized the conventional process ways of network pharmacology..." have changed to "Additionally, we optimized the conventional methodologies of network pharmacology...".

“A total of 85 absorbed constituents were identified or tentatively characterized in dosed plasma, including 32 components in Radix Scutellaria, 27 components in Licorice and 65 components in RSLDP” have changed “A total of 85 absorbed constituents were identified or tentatively characterized in dosed plasma, including 32 components in the group of Radix Scutellaria, 27 components in the group of Licorice, and 65 components in the group of RSLDP”. 

Comment 2: There is a lack of context and background information in the results and discussion section. It would be helpful for the authors to better explain the rationale behind the study, the significance of the findings, and how they relate to existing knowledge in the field.

Reply: We sincerely appreciate the valuable comments. We have added more context and background information in the results and discussion section of the revised manuscript.

Comment 3: The manuscript would benefit from a more detailed discussion of the limitations of the study, as well as suggestions for future research. Section 3, which is titled “results and discussion”, does not adequately discuss the gap between the results and the literature. Only four pieces of literature are cited in this section, which is unusual for a discussion section.

Reply: Thanks a lot for your meaningful comments. We have fully described the limitations and prospects of the discussion section. And we have adequately discussed the relationship between the experimental results and the references in the Results and Discussion section of Section 3. We have added more references to support our idea in the revised manuscript.

Comment 4: The figures should be clear and of high quality.

e.g. Figure 9 has been stretched strangely. Some gene names are not fully displayed or difficult to read.

Reply: Thank you very much for your comments. The quality and clarity of Figure 9 has been improved.

Comment 5: The format of references is seriously inconsistent. The references should be formatted consistently, and the authors should ensure that all cited literature is relevant and up-to-date.

Reply: Thanks a lot for your comments. We are sorry for our carelessness. Based on your comments, we have made the corrections to make the formatting of references consistent. And we have ensured that all literature cited is relevant and up-to-date.

---

## [Editor Report · Decision Letter 1]

28 Jun 2023

PONE-D-23-05859R1Systematically uncovering the absorbed effective substances of Radix scutellaria-licorice drug pair in rat plasma against COVID-19 using a combined UHPLC-Q-TOF-MS analysis and target network pharmacologyPLOS ONE

Dear Dr. yiran,

Thank you for submitting your manuscript to PLOS ONE. After careful consideration, we feel that it has merit but does not fully meet PLOS ONE’s publication criteria as it currently stands. Therefore, we invite you to submit a revised version of the manuscript that addresses the points raised during the review process.

The author has made serious revisions to the manuscript, which has greatly improved the quality of the article. However, there is one small issue that I would like to discuss with you:

The acknowledgment section does not require thanking the reviewer; Thank you to the laboratory personnel and suggest specific names; Co authors should not be reflected in the acknowledgment section. Suggest adding information such as funding to support this research.

We look forward to receiving your revised manuscript.

Kind regards,

Chun-Hua Wang

Academic Editor

PLOS ONE

Journal Requirements:

Additional Staff Editor Comments: PLOS ONE requires that published manuscripts use language which is 'clear, correct, and unambiguous', see our criteria for publication at https://journals.plos.org/plosone/s/criteria-for-publication#loc-5. We therefore request that you revise the text to fix the grammatical errors and improve the overall readability of the text.

We suggest you have a fluent English-language speaker thoroughly copyedit your manuscript for language usage, spelling, and grammar. If you do not know anyone who can do this, you may wish to consider employing a professional scientific editing service.

---

## [Author Response · Author response to Decision Letter 1]

3 Jul 2023

Editor #1:

Comment 1: The acknowledgment section does not require thanking the reviewer; Thank you to the laboratory personnel and suggest specific names; Co authors should not be reflected in the acknowledgment section. Suggest adding information such as funding to support this research.

Reply: Thanks for your comments. We have revised the acknowledgment section of the revised manuscript. We have added funding sources to the funding information section of the submission system.

Editor #2:

Comment 1: We therefore request that you revise the text to fix the grammatical errors and improve the overall readability of the text. We suggest you have a fluent English-language speaker thoroughly copyedit your manuscript for language usage, spelling, and grammar. If you do not know anyone who can do this, you may wish to consider employing a professional scientific editing service.

Reply: Thank you very much for your helpful suggestion. We have tried our best to revise the grammar, spelling, and language usage of the entire manuscript.

Journal Requirements #3: 

Comment 1: Please review your reference list to ensure that it is complete and correct.

Reply: Thanks a lot for your meaningful comments. We have ensured that all literature cited is complete and correct.

---

## [Editor Report · Decision Letter 2]

12 Jul 2023

Systematically uncovering the absorbed effective substances of Radix scutellaria-licorice drug pair in rat plasma against COVID-19 using a combined UHPLC-Q-TOF-MS analysis and target network pharmacology

PONE-D-23-05859R2

Dear Dr. yiran,

We’re pleased to inform you that your manuscript has been judged scientifically suitable for publication and will be formally accepted for publication once it meets all outstanding technical requirements.

Kind regards,

Chun-Hua Wang

Academic Editor

PLOS ONE

---

## [Editor Report · Acceptance letter]

1 Aug 2023

PONE-D-23-05859R2 

Systematically uncovering the absorbed effective substances of Radix Scutellaria-licorice drug pair in rat plasma against COVID-19 using a combined UHPLC-Q-TOF-MS analysis and target network pharmacology 

Dear Dr. Jin:

I'm pleased to inform you that your manuscript has been deemed suitable for publication in PLOS ONE. Congratulations! Your manuscript is now with our production department. 

Kind regards, 

on behalf of

Dr. Chun-Hua Wang 

Academic Editor

PLOS ONE